# ROYAL SOCIETY
# OPEN SCIENCE

analytical chemistry

direct mass spectrometry, nanospray electrospray ionization, nanospray capillary, metronidazole, human urine, 1, 3, 6- polytyrosine

**Author for correspondence:**
Sara Amer
e-mail: sara.m.amer@hotmail.com

---

†Deceased 3 March 2017.

This article has been edited by the Royal Society of Chemistry, including the commissioning, peer review process and editorial aspects up to the point of acceptance.

# Direct nano-electrospray ionization tandem mass spectrometry for the quantification and identification of metronidazole in its dosage form and human urine

Sara Amer[1,2], Walaa Zarad[1], Heba El-Gendy[1], Randa Abdel-Salam[3], Ghada Hadad[3], Tsutomu Masujima[2,†] and Samy Emara[1]

[1]Faculty of Pharmacy, Misr International University, Km 28 Ismailia Road, Cairo 11865, Egypt
[2]Quantitative Biology Center (QBiC), RIKEN, 6-2-3 Furuedai, Suita, Osaka 565-0874, Japan
[3]Pharmaceutical Analytical Chemistry Department, Faculty of Pharmacy, University of Suez Canal, Ismailia 41522, Egypt

SA, 0000-0002-7381-5361

A rapid, sensitive and direct nano-electrospray ionization-tandem mass spectrometry (NS-ESI-MS/MS) method, using an offline nanospray (NS) capillary, has been developed and validated for the analysis of metronidazole (MTZ). A mixture of 2 µl MTZ sample solution prepared in an ionization solvent consisting of methanol : water : formic acid in a ratio of 80 : 20 : 0.3, together with 2 µl of an internal standard (IS), 1,3,6-polytyrosine, is loaded into the back of the NS capillary. The NS capillary was fitted into the ion source at a distance of 3 mm between the NS tip and MS orifice. The sample is then analysed and acquired a sustainable signal that allowed for data compilation across various data points for MTZ identification and quantification. The quantification relied on the ratio of the $[M + H]^+$ peaks of MTZ and IS with $m/z$ values of 172.0717 and 182.0812, respectively, while the identification relied on the MS/MS of the precursor ions $[M + H]^+$ of both compounds and their fragments at 128.05 for MTZ and 165.1 and 136.07 for the IS. The NS-ESI-MS/MS method was accurate and precise for the quantification of MTZ over the concentration range from 2.5 to 25 000 ng ml$^{-1}$.

The applicability of the method was confirmed by MTZ analysis in its pharmaceutical dosage form and detection of the analyte in clinical human urine samples without any sample treatment procedure.

# 1. Introduction

Metronidazole (1-(2-hydroxyethyl)-2-methyl-5-nitromidazole) (MTZ) is one of the essential drugs used to treat protozoal infections of *Trichomonas vaginalis* and anaerobic microbial infections [1]. MTZ is also used to treat infections such as giardia in pets [2]. It is the most widely used nitroimidazole, and is well tolerated at doses of less than or equal to 2 g d$^{-1}$. It is an essential drug in the Egyptian market due to the high level of pollution and other factors that contribute to a high number of cases involving diarrhoea and intestinal infections [3].

Several methods have been reported for the determination of MTZ from its individual or combined formulations with other active ingredients using spectrophotometry [4–7], electrochemistry [8–11], capillary electrophoresis [12–14], gas chromatography (GC) [15–18], high-performance liquid chromatography (HPLC) coupled with ultraviolet [19–25] and mass spectrometric detector methods [26–34].

The ability for separation has allowed LC to be the first choice in method selection for a wide range of drugs. Perhaps the greatest advantage in using LC as a separation technique in coupling with MS as a detector (LC-MS) is the ability to identify multiple compounds in unresolved chromatographic peaks.

However, chromatographic methods have their drawbacks. All previous published LC-MS methods clarified that mobile phase composition and type of stationary phase have to be investigated to understand the occurrence of both analyte and internal standard (IS) peaks. This usually involved prolonged chromatography run time. Also, carry-over is one of the most common problems in developing the LC-MS method and the sources of this problem range from hardware devices and the selection of appropriate rinse solvents to challenges with chromatography. Choosing interface and mass spectrometer settings is another issue that should be considered. All these have been considered due to the desire to set a shorter timeline in drug discovery and development which led to the need for rapid and sensitive approaches for drug evaluation in dosage forms, as well as monitoring in biological matrices for further study.

The nanospray (NS) capillary, produced by Humanix (Hiroshima, Japan), devised a new coupling technique with MS. Masujima's group has been using NS capillaries in the field they have coined as 'live single cell MS', in which they trap single cells for analysis in order to identify many bioactive compounds [35–42]. However, with continuous improvement, these NS capillaries can stand out as a promising analytical tool for pharmaceutical quality control applications. The current study uses these NS capillaries as a replacement for LC-MS interface systems and directly sprays the sample into the mass spectrometer. Their unique shape and minute tip bore size allow for optimum electrospray and small droplet size, leading to more ion formation. Also, the instrument's unique set-up under ambient conditions can be applied on a wide variety of thermo-sensitive drugs. In the present study, we did not use the LC interface and its associated problems. The use of disposable NS capillaries reduces the potential for contamination, carry-over effect. In addition, the small volume approach allows for the introduction of less matrix components into the MS system, thus extending the useful life of the equipment, in contrast with traditional LC methods. This study explores the use of these NS capillaries in the development and validation of a direct NS-ESI-MS/MS method to evaluate the analysis of MTZ in tablet dosage forms in 30 s, with the intention of possible applications for various routine pharmaceutical developments. To the best of our knowledge, it is the first study of a drug analysis using direct NS-ESI-MS/MS. Sample solutions were loaded into the NS capillaries and then analysed using this technique, acquiring sustainable signal that allowed for swift data compilation across various data points for compound identification and quantification. The detection of MTZ in clinical urine samples is still required and will reveal the applicability of the direct NS-ESI-MS/MS design to significantly reduce sample run time, drug detection limit and may also have an impact on the qualitative and quantitative analysis of drugs in biological fluids. The applicability of the proposed direct NS-ESI-MS/MS method was assessed by analysing random urine samples collected from patients in treatment with MTZ.

# 2. Material and methods

## 2.1. Chemical and materials

HPLC grade methanol (Tokyo Chemical Industry Co., LTD, Japan) was used. Formic acid (Sigma Aldrich, Germany) was of reagent grade (greater than or equal to 95%). 1,3,6- polytyrosine solution (Thermo Fisher Scientific) was used as IS. MTZ was a gift from Sanofi-Aventis (Egypt). Flagyl tablets (Batch no. 5EG102) were manufactured by Sanofi-Aventis. Each tablet contains 500 mg MTZ. Ultrapure water (Organo, Puric-ω, Tokyo, Japan) was prepared daily and used throughout the experiment.

## 2.2. Standard solutions and calibration

An equivalent amount of MTZ powder was dissolved in methanol and used as a stock solution of 5 mg ml$^{-1}$. From the stock solution, serial dilutions were prepared to cover the concentration range of 0.25–2500 µg ml$^{-1}$ of working solutions using the same solvent. All stock and working solutions were kept at −20°C. The calibration curve of MTZ was prepared by diluting the above working solutions 100 times with methanol to obtain a series of calibration standard solutions ranging from 2.5 to 25 000 ng ml$^{-1}$. Each concentration was injected in replicates of three.

## 2.3. Instrumentation

Full MS analysis was carried out on a high-resolution mass spectrometer (OrbitrapVelos Pro, Thermo Fisher Scientific). The Velos Pro mass spectrometer was equipped with an offline NS ion source (Thermo Fisher Scientific). The NS ion source was positioned on a manual XYZ stage, and the XYZ stage was placed in front of the mass spectrometer inlet. Once the NS capillary was mounted on the ion source, the XYZ stage was then used to move the NS capillary to the optimum position. To aid in this, a video camera was positioned between the NS capillary tip and the inlet of the MS instrument, allowing for the user to observe and calculate the distance between the two. MS/MS analysis was done in the ion trap mode of the instrument. Helium gas (99.999% purity) was used as the collision gas. Collision energy was set between 20 and 25 eV in collision-induced dissociation (CID) mode. The ionization solvent used consisted of 80 : 20 : 0.3 ratio between methanol : water : formic acid. Solutions to be analysed were loaded into NS capillaries of 1 µm tip pore size (Humanix, Hiroshima, Japan) using pipette tips (Eppendorf, GELoader tips). The spray voltage applied was 1000 V. Full MS scans were monitored in the following ranges: 150–380, 150–212, 165–190 and alternating scans between 170–174 and 180–184, each for 30 s. All MS/MS spectra were recorded using an average across 30 s of scans. Ion trap calibration was done once at the very beginning of the experiment in the positive mode using the positive ion mode calibration solution, which was prepared as stated in the Thermo Scientific LTQ Velos Pro Manual. Positive mode Fourier transform (FT) manual calibration was done every morning with the NS ionization source using commercially available polytyrosine solution.

## 2.4. Dosage form analysis

According to the label, Flagyl contains 500 mg of MTZ per tablet (Sanofi-Aventis Egypt). A total of 10 tablets were weighed and then pulverized and homogenized. An accurately weighed portion of the powder equivalent to 250 mg of MTZ was transferred to a 50 ml volumetric flask and then dissolved in about 30 ml of methanol using a shaker for 30 min. After extraction, the obtained solution was quantitatively diluted to volume with methanol. The solution was then filtered; the first portion of the filtrate was discarded and the remainder was used as stock sample solution. A known volume of sample stock solution was diluted quantitatively with methanol to obtain a concentration of 500 ng ml$^{-1}$ MTZ.

## 2.5. Urine analysis

The development of a convenient direct MS tool to detect MTZ in urine samples is still required. Consequently, the developed direct NS-ESI-MS/MS method was further applied to detect MTZ in random human urine samples with high sensitivity, selectivity and without any sample pretreatment or separation procedures. Urine samples were collected from volunteers after the oral administration of a single dose of MTZ in some cases. In other cases, urine samples were collected after the

administration of MTZ and other prescribed drugs. Urine samples were loaded into the NS capillaries and processed as mentioned under instrumentation.

## 2.6. Selectivity

Five blank preparations (ionization solvent) were prepared and injected to determine the selectivity of the method. The degree of selectivity was determined by the presence or absence of interference $m/z$ peaks at the $m/z$ values of MTZ and IS.

## 2.7. Data analysis

Data acquisition was performed using Thermo Xcalibur software and raw data were analysed on Microsoft Excel. Detected peaks were estimated by the exact mass value of ±5 ppm tolerance, and annotated by KEGG database (http://www.kegg.jp/kegg/) and HMDB database (http://www.hmdb.ca/). MS/MS spectra were compared against MetFrag (https://msbi.ipb-halle.de/MetFrag/).

## 2.8. Assay validation

Three different concentration levels (10, 100 and 10 000 ng ml$^{-1}$) of MTZ were prepared to determine intra- and inter-day precision and the accuracy of the proposed method against the calibration curve. Intra-assay precision was assessed as the relative standard deviation (RSD) of the mean MTZ/IS ratio obtained on the same day. Inter-assay precision was calculated using the RSD of the mean MTZ/IS ratio on five consecutive days. Accuracy was determined as the per cent of the relative errors (REs) of the mean measured concentrations using the following equation:

$$RE\% = \left[ \frac{\text{mean measured concentration} - \text{nominal concentration}}{\text{nominal concentration}} \right] \times 100.$$

# 3. Results and discussion

## 3.1. NS offline ionization advantages

The NS capillaries are minute, hollow, borosilicate glass tubes 28 mm long and, depending on the selected size, have a tip bore size ranging from 1 to 10 μm wide. These tips are platinum coated, allowing for electrical conductivity [43]. Samples are loaded into the NS capillary from the back side using a gel-loading pipette. After mounting the capillary onto the ion source, electric voltage is applied to start the spray, producing an electric field between the NS capillary and MS inlet. Under this influence, ions of similar polarity, as the applied voltage, are repelled and migrate towards the NS tip.

At the moment of electric voltage application, ions of similar charge to that of the applied voltage are repelled and migrate towards the NS outlet. In the current study, the positive charge of the applied voltage repels the ions of similar charge, causing them to migrate towards the NS outlet. A Taylor cone is then formed due to the equilibrium between the electrostatic force and the surface tension of the sample liquid. While the electrostatic force pulls the liquid from the capillary towards the MS inlet, the surface tension runs in the opposite direction. Immediately after, a fine jet of the solution is released from the tip of the Taylor cone, which then easily disperses into small droplets. These droplets are composed of equidistant multicharges and many analyte molecules. A decreased size of these droplets initially occurs due to solvent evaporation, until repulsion forces (Coulomb forces) of the like charges overcome the surface tension of the said droplet. At this point, the Rayleigh limit is said to be reached and Coulomb fission occurs. This process is repeated until singular charged analyte molecules are produced, which the MS inlet enters. This process is illustrated in figure 1.

Due to the inherently low flow rates achieved solely by electrostatic and capillary forces, nano-ESI has several advantages over traditional ESI techniques with higher flow rates. A lower flow rate solution is known to reduce the size of primarily produced charged droplets, and the generated droplets by an NS tip are about 100 times smaller than those of conventional ESI sources [44], allowing for a more efficient use of the total sample with minimum loss of analyte material in comparison with conventional ESI techniques. Therefore, initial droplets need fewer droplet fission events and less solvent evaporation prior to the release of a gas-phase ion. Analytically, the developed direct NS-ESI-MS/MS method is

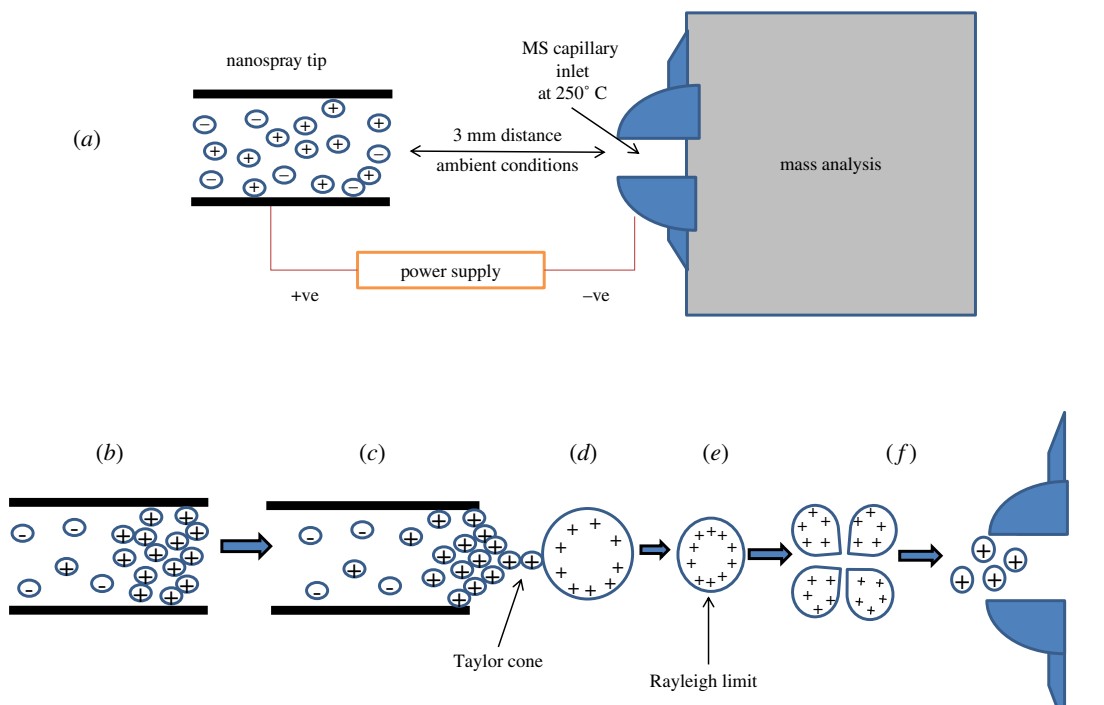

**Figure 1.** (*a*) The overall framework of NS ionization. (*b*) Upon application of positive voltage, ions that carry positive charge are repelled and migrate to the NS tip. (*c*) The positive ions continue to accumulate until a Taylor cone is produced. (*d*) The jet stream produced from the Taylor cone produces comparatively large primary multicharged droplets. (*e*) Desolvation decreases the size of the primary droplets, pushing the like charges closer and closer until the Rayleigh limit is reached, in which repulsion forces surpass surface tension and (*f*) Coulomb fission occurs. This process is then repeated until singularly charged analyte molecules are formed, which are analysed by the mass spectrometer.

unique, in that it achieves a quantifiable signal without the need for separation, thereby removing the downsides of method coupling.

## 3.2. Method optimization

### 3.2.1. Voltage

For MTZ determination, preliminary experiments were carried out on a voltage range of 800–1300 V. It was observed that the applied low voltage caused considerable instability in the analyte signal, whereas the voltage greater than 1000 V led to the loss of the MS signal. An adequate voltage application of 1000 V, resulting in excellent sensitivity and stability of the analyte signal, was selected for further study.

### 3.2.2. MS inlet temperature

Different inlet temperatures were tested to promote thermal declustering, if needed. For this experiment, a range of 200–400°C was evaluated in increments of 50. Little difference was shown between the results, indicating minimal ion clustering of the sample, so a default of 250°C was set.

### 3.2.3. Distance to MS inlet

Experimentation was done on a singular concentration at different distances between the NS capillary tip and MS inlet ranging 1–4 mm, measured by the scaled XYZ stage and video camera in order to find the position that attained the highest ion count. The best results were achieved at a distance of 3 mm, which was kept constant throughout the present study.

### 3.2.4. Ionization solvent, composition and ratio

The effect of ionization solvent for offline ionization of MTZ was done with different ratios between methanol and ultrapure water, with the addition of 3% formic acid. A 100 ng ml$^{-1}$ solution of standard

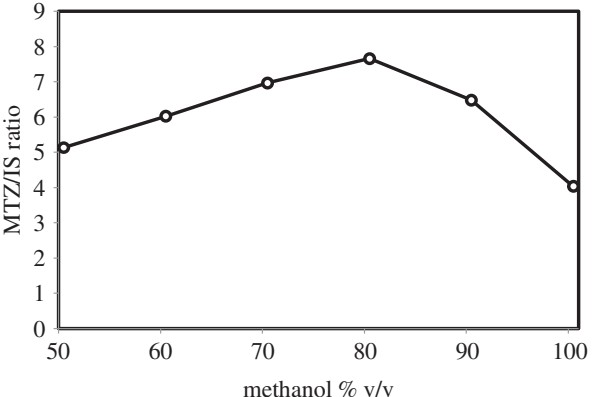

**Figure 2.** Dependence of the signal intensity of MTZ on the methanol ratio of the ionization solvent.

**Figure 3.** Comparison between IS and MTZ signal at (*a*) 10 ng ml$^{-1}$, (*b*) 100 ng ml$^{-1}$ and (*c*) 10 000 ng ml$^{-1}$ over a 30 s interval.

MTZ was prepared and injected with different solvent ratios, ranging from 50–100%, in replicates of 3. Average [M + H]$^{+}$ signal intensities of MTZ to IS were analysed and summarized in figure 2. As shown, the maximum ionization of MTZ was observed at 80% methanol, as indicated by the signal

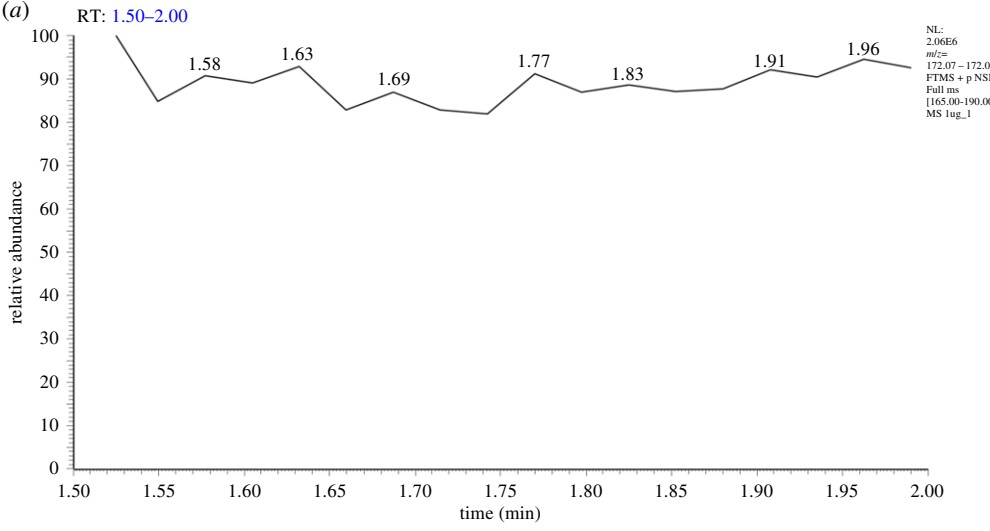

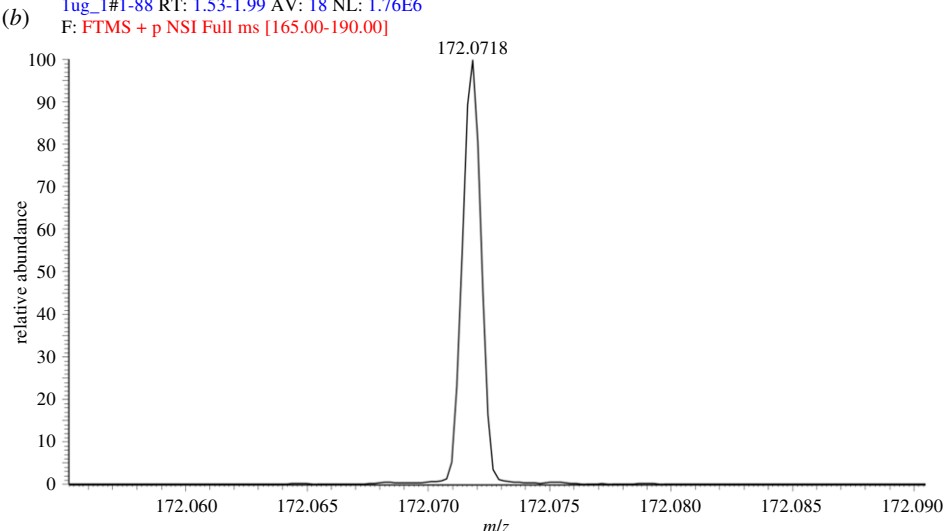

**Figure 4.** Screenshot of XCalibur platform depicting TIC (*a*) of selected MTZ *m/z* peak (*b*).

ratio of analyte to IS. Accordingly, the optimum ionization solvent for MTZ was evaluated to be 80 : 20 : 0.3 (v/v) methanol : water : formic acid.

### 3.2.5. Internal standard selection

The application of an IS was the key in turning direct MS from a qualitative to a quantitative tool. Most, if not all, methods choose their IS to either be a compound of very similar chemical structure or the labelled isotope of the compound to be analysed. This is to ensure both the drug and IS undergo the same conditions, and if there is a loss, both are lost with the same ratio. For these reasons, deuterium-labelled isotopes are considered the optimum choice for mass spectrometric analysis as they are almost identical to the tested compound except that 4–6 hydrogen atoms are replaced with deuterium [45,46]. But labelled isotopes are much more expensive than their unlabelled counterparts, and some may require special storage conditions in order to maintain their integrity.

By contrast, the current method does not require a chemically related IS, because there is no pretreatment step nor chromatographic separation in which there is any chance of loss of sample. The only requirement in the choice of an IS is that it be of adequate, stable and similar ionization efficiencies. For this reason, the selected IS was the same solution used for FT calibration-1,3,6-polytyrosine solution. This solution has excellent ionization, and contains three major *m/z* peaks, allowing for the quantification at any *m/z* range. This also gives us the advantage of quantifying several compounds within the same run. This undergoing experiment used the peak of 1-tyrosine as

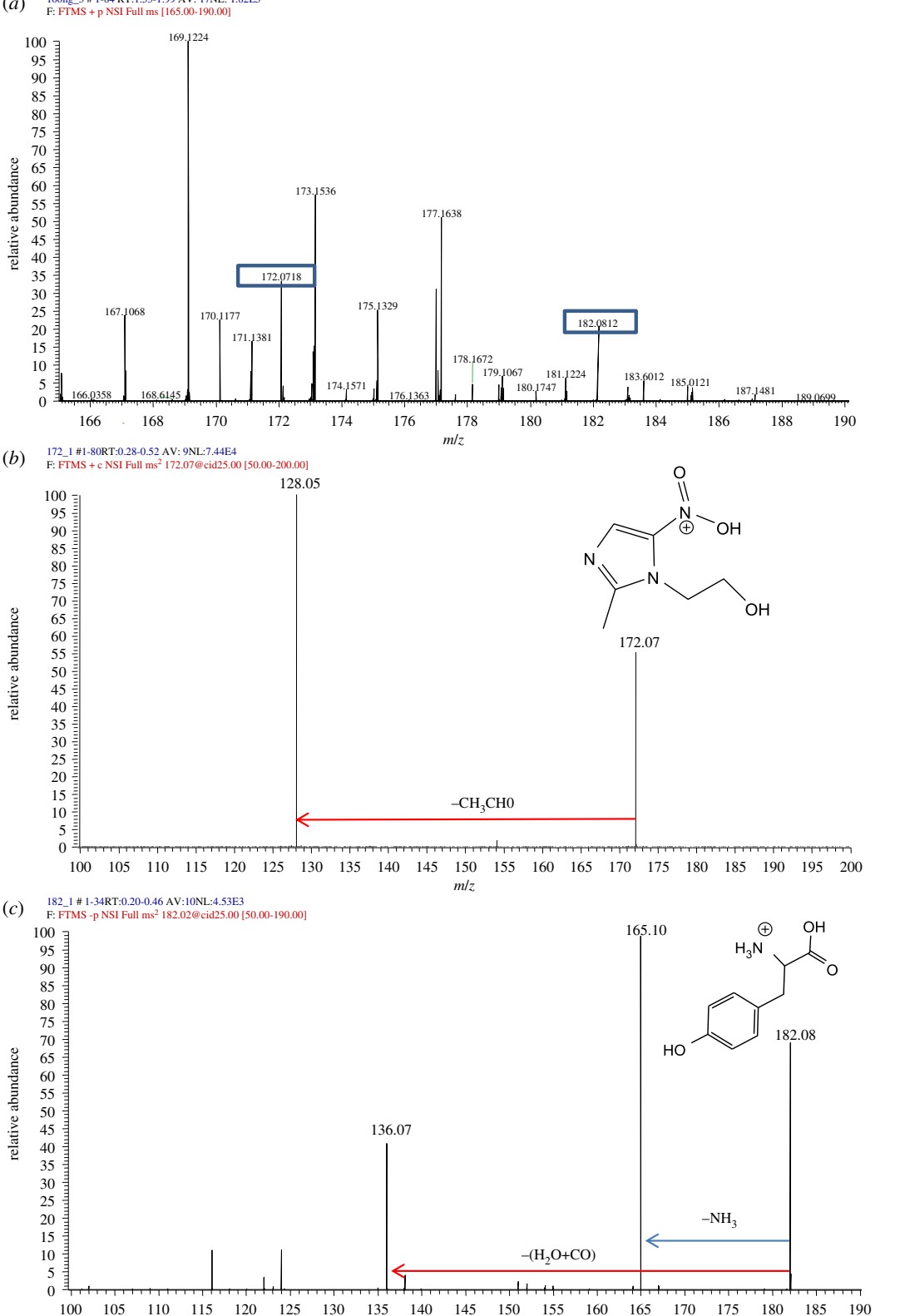

**Figure 5.** (*a*) Full mass spectrum depicting both *m/z* = 172.0717 of MTZ and *m/z* = 182.0795 of IS. (*b*) MS/MS spectrum of MTZ at CID = 25 eV. (*c*) MS/MS spectrum of 1-tyrosine at CID = 25 eV.

the IS, as it is the closest in *m/z* value to that of MTZ. To ensure our hypothesis, a comparison between MTZ and IS signal was assessed at three different concentration levels, and the results are shown in figure 3. This graphical depiction illustrates that there is a clear correlation between the ionization

**Table 1.** Summary of calibration results of MTZ over three consecutive days by direct NS-ESI-MS/MS method.

| calibration range (2.5–25 000 ng ml$^{-1}$) | day 1 | day 2 | day 3 | average |
| --- | --- | --- | --- | --- |
| slope | 0.0678 | 0.0603 | 0.0657 | 0.0646 |
| intercept | −0.4294 | −0.0507 | 0.4611 | −0.0063 |
| $R^2$ | 0.9982 | 0.9959 | 0.9977 | 0.9973 |

efficiencies of the two signals, leading to the conclusion that both signals of the analyte and IS are continuously affected at the same ratio.

### 3.2.6. Sample loading

An aliquot 2 µl sample solution, together with 2 µl of an IS, is loaded into the back of the NS capillary. It is important to note that any handling of an NS capillary must be carefully done using forceps. Doing so will prevent any damage to the outer coat, prevent contamination by the human hand as well as greatly decrease the chance of breaking the NS capillary tip. After the solvent has been added, the NS capillary is to be centrifuged for a couple of seconds. This step is critical in removing the air gap at the front end of the NS capillary and moving the sample to the very tip, for maximum spray results. This way, the NS capillary will be ready to be mounted within the offline ion source.

## 3.3. Data extraction and interpretation

The data were extracted from XCalibur to Microsoft Excel in the following steps: (i) two active windows in XCalibur should be open—one depicting the total ion chromatogram (TIC) of the run, depicted in figure 4a and the other representing the average mass spectrum of the run, depicted in figure 4b. (ii) Activate the mass spectrum window by clicking on the pin icon on the top right corner. Zoom in on $m/z = 172.0717$ within the mass spectrum. (iii) Activate the TIC window by clicking on its respective pin icon. Then select the $m/z$ from 172.07 to 172.074. (Note: a very narrow range was chosen due to the fact that as concentration increased, the base peak width increased. Also, as a limitation of XCalibur, it is not possible to only choose one exact $m/z$. If one does so, the programme automatically calculates the TIC of this $m/z \pm 1$.) A stable signal should appear in the TIC window. This indicates that the ionization of MTZ is stable and constant throughout the run. (iv) Right click the TIC window and choose 'Export@Clipboard (Chromatogram)'. Then, open an empty Microsoft Excel file. Choose the first cell and paste the contents. The TIC window will be pasted and numerical points in a XY format, X as time and Y as peak intensity. (v) Below the column of the MTZ intensity, calculate the area under the curve (AUC) of the TIC using the following equation:

$$= \text{Sum product} \left( (\text{A2:A100} - \text{A1:A99}) \times (\text{B2:B100} + \text{B1:B99}) \right) \times 0.5$$

This equation represents the trapezoid rule, which divides the graph into multiple trapezoids and calculates the summation of all trapezoids formed. Note: A1 and B1 represent the cell position that contains the first XY point; A2 and B2 represent the position of the second XY point; A99 and B99 represent the position of the second to last XY point; and A100 and B100 represent the position of the last XY point. (vi) Repeat the previous steps on the same run, but on the peak of the IS at $m/z$ range of 82.079–182.084. Paste the results next to the results of MTZ and calculate the AUC. (vii) Finally, calculate the ratio of MTZ/IS as $\text{AUC}_{\text{MTZ}}/\text{AUC}_{\text{IS}}$.

## 3.4. Quantification and confirmation

Full MS spectra were used for quantification in a range containing both $m/z$ values of $[M + H]^+$ of MTZ (172.0717) and IS (182.0812), both seen in the spectrum of figure 5a. MS/MS analysis was performed for confirmation of both drug and IS and were compared to the accepted spectral patterns of the respective compounds. As illustrated in figure 5b, the precursor of ion $[M + H]^+$ at $m/z = 172.1$ gave way to 2-methyl-5-nitromidazole after the complete removal of 1N side chain at $m/z = 128.05$. Figure 5c depicts the MS/MS

**Table 2.** Precision and accuracy validation of MTZ by direct NS-ESI-MS/MS method.

| concentration | 10 ng ml$^{-1}$ | | | | |
|---|---|---|---|---|---|
| run no. | day 1 | day 2 | day 3 | day 4 | day 5 |
| 1 | 113.79 | 101.09 | 108.62 | 92.36 | 102.87 |
| 2 | 106.73 | 111.73 | 97.12 | 99.69 | 106.08 |
| 3 | 108.63 | 95.99 | 96.26 | 111.11 | 120.34 |
| 4 | 112.07 | 94.31 | 107.05 | 110.60 | 111.31 |
| 5 | 96.65 | 117.04 | 106.22 | 103.62 | 107.12 |
| | intra-day results | | | | |
| mean | 107.57 | 104.03 | 103.05 | 103.48 | 109.54 |
| s.d. % | 6.70 | 9.95 | 5.88 | 7.85 | 6.74 |
| RSD % | 6.23 | 9.56 | 5.70 | 7.59 | 6.15 |
| RE % | 7.57 | 6.03 | 3.05 | 3.48 | 9.54 |
| | inter-day results | | | | |
| mean | 105.54 | | | | |
| s.d. % | 2.86 | | | | |
| RSD % | 2.71 | | | | |
| RE % | 5.94 | | | | |

| concentration | 100 ng ml$^{-1}$ | | | | |
|---|---|---|---|---|---|
| run no. | day 1 | day 2 | day 3 | day 4 | day 5 |
| 1 | 115.60 | 117.03 | 106.02 | 103.24 | 110.28 |
| 2 | 111.70 | 112.82 | 97.96 | 100.40 | 107.57 |
| 3 | 120.43 | 111.75 | 102.09 | 96.49 | 106.61 |
| 4 | 116.52 | 110.06 | 107.07 | 105.73 | 114.13 |
| 5 | 113.65 | 108.55 | 112.45 | 112.23 | 116.95 |
| | intra-day results | | | | |
| mean | 115.58 | 112.04 | 105.12 | 103.62 | 111.11 |
| s.d. % | 3.28 | 3.22 | 5.45 | 5.91 | 4.38 |
| RSD % | 2.84 | 2.88 | 5.18 | 5.70 | 3.94 |
| RE % | 15.58 | 12.04 | 5.12 | 3.62 | 11.11 |
| | inter-day results | | | | |
| mean | 109.49 | | | | |
| s.d. % | 4.99 | | | | |
| RSD % | 4.56 | | | | |
| RE % | 9.49 | | | | |

| concentration | 10 000 ng ml$^{-1}$ | | | | |
|---|---|---|---|---|---|
| run no. | day 1 | day 2 | day 3 | day 4 | day 5 |
| 1 | 95.55 | 105.28 | 99.16 | 94.41 | 110.28 |
| 2 | 101.37 | 107.32 | 101.22 | 103.53 | 111.13 |
| 3 | 91.20 | 104.56 | 99.72 | 99.56 | 111.46 |
| 4 | 96.05 | 106.84 | 103.11 | 94.78 | 105.87 |
| 5 | 105.74 | 109.25 | 107.49 | 98.37 | 107.45 |

(*Continued.*)

**Table 2.** (*Continued.*)

| concentration | 10 000 ng ml$^{-1}$ | | | | |
|---|---|---|---|---|---|
| run no. | day 1 | day 2 | day 3 | day 4 | day 5 |
| mean | 97.98 | 102.14 | 98.13 | 109.23 | 106.65 |
| s.d. % | 5.64 | 1.84 | 3.35 | 3.75 | 2.45 |
| RSD % | 5.75 | 1.72 | 3.28 | 3.82 | 2.25 |
| RE % | −2.01 | 6.65 | 2.14 | −1.86 | 9.23 |
| mean | 102.83 | | | | |
| s.d. % | 5.04 | | | | |
| RSD % | 4.90 | | | | |
| RE % | 2.83 | | | | |

**Table 3.** Determination of MTZ in commercial formulations through 500 ng ml$^{-1}$ sample solution by direct NS-ESI-MS/MS method.

| run no. | 1 | 2 | 3 | 4 | 5 | average |
|---|---|---|---|---|---|---|
| concentration found (ng ml$^{-1}$) | 518.21 | 545.45 | 538.33 | 510.16 | 490.65 | 520.56 |
| recovery % | 103.64 | 109.09 | 107.67 | 102.03 | 98.13 | 104.11 |
| RSD % | 2.58 | 6.43 | 5.42 | 1.44 | 1.32 | 3.44 |
| RE % | 3.64 | 9.09 | 7.67 | 2.03 | −1.87 | 4.11 |

spectrum of 1-tyrosine in which the precursor ion, $[M + H]^+$ at $m/z = 182.1$ is dissociated into several fragments including the loss of $NH_3$ at $m/z = 165.1$, and the loss of $H_2O + CO$ at $m/z = 136.07$.

## 3.5. Validation

### 3.5.1. Linearity

The effect of $m/z$ range was examined in order to achieve the most accurate and precise linear equation. The method compared the AUC of the MTZ/IS ratio versus concentration over four $m/z$ ranges: 150–380, 150–212, 165–190 and alternating scans between $m/z$ ranges 170–174 and 180–184, each run for 30 s. Regression equations were compared and the $R^2$ value ($R^2$) of each range was found to be 0.979, 0.990, 0.999 and 0.993, respectively. These results indicate that a smaller range that contains both analyte peak and IS peak gives more accurate results. We inferred also that too small a range may have too low an ion count to be reliable, and that both analyte and IS should be present in the same scan to undergo the exact same instrument conditions. Therefore, the range from $m/z$ 165 to 190 was chosen as the best range to be used for quantification. Once $m/z$ range was established, day-by-day calibration curve over the concentration range from 2.5 to 25 000 ng ml$^{-1}$ was done on 3 different days in order to observe the degree of change in regression. Over 3 days, $R^2$ was calculated to be 0.9977, 0.9959 and 0.9982, respectively ($n = 3$) (table 1).

### 3.5.2. Limit of detection and limit of quantitation

The limit of detection (LOD) and limit of quantification (LOQ) were determined according to the ICH guidelines for validation of analytical procedures [47] and were found to be 0.5 ng ml$^{-1}$ and 2.5 ng ml$^{-1}$, respectively.

### 3.5.3. Accuracy and precision

Intra- and inter-day precisions and accuracies of the three different concentration levels are summarized in table 2. Daily recovery per cent ranged from 91.21 to 120.44%. Intra-day RSD% ranged from 1.72 to

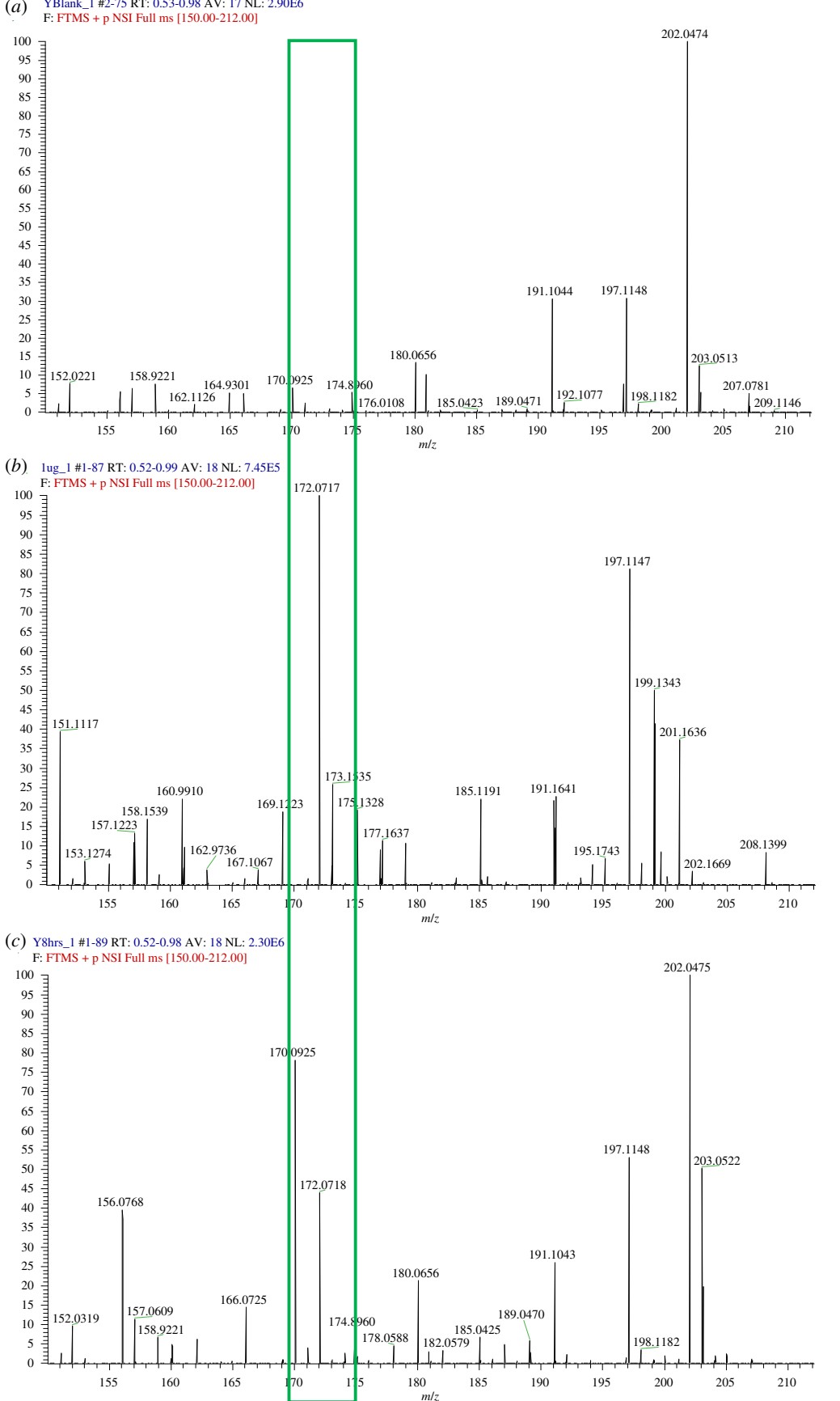

**Figure 6.** Depicting spectrum of (*a*) drug-free human urine sample, (*b*) drug-free urine spiked with 50 ng ml⁻¹ MTZ and (*c*) clinical urine sample after oral administration of MTZ.

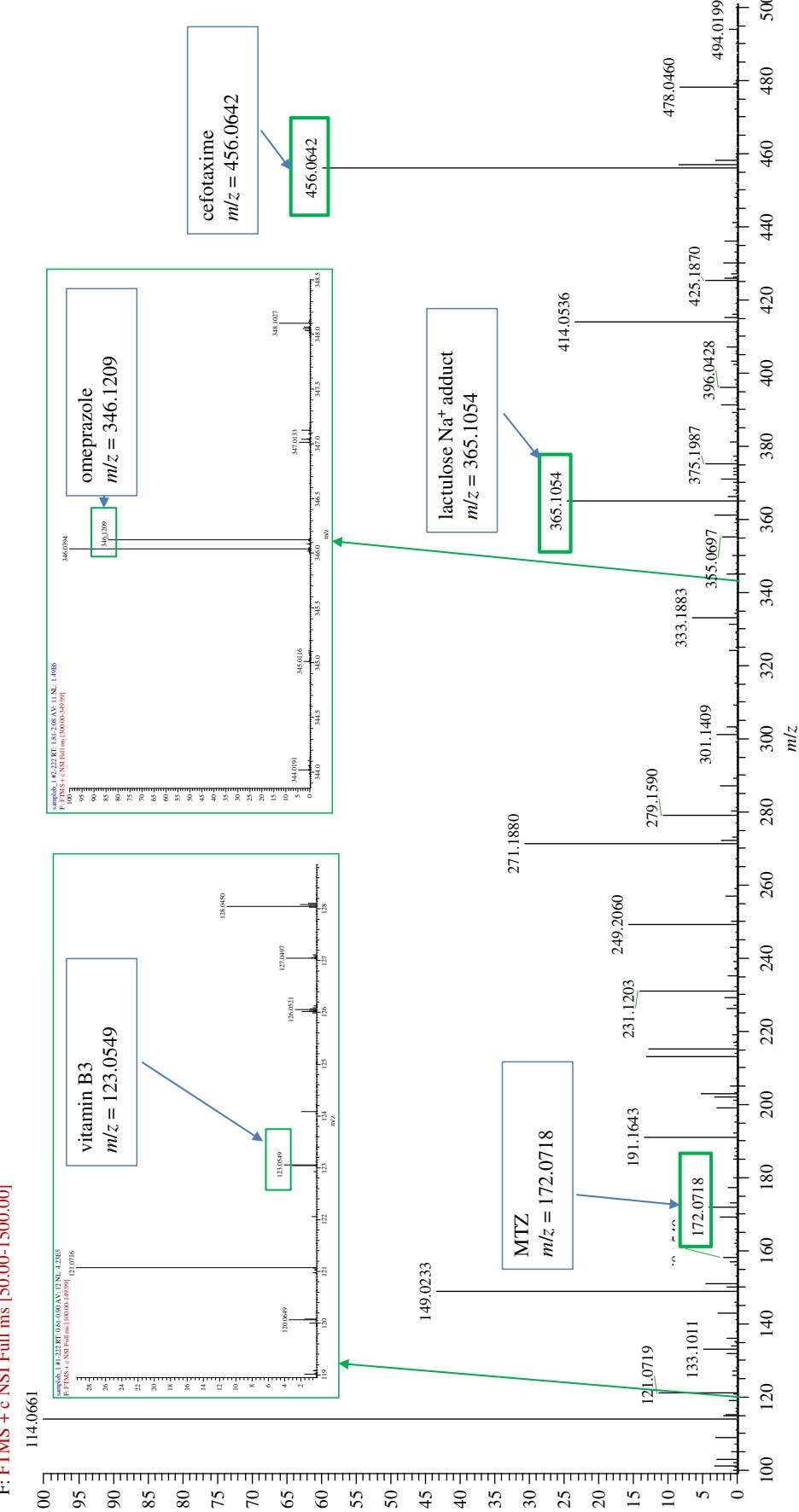

**Figure 7.** A mass spectrum of a clinical urine sample taken from a patient after oral ingestion of a number of drugs, including vitamin B3 ($m/z$ = 123.0549), MTZ ($m/z$ = 172.0718), omeprazole ($m/z$ = 346.1209), lactulose ($m/z$ = 365.1064 Na$^+$ adduct) and cefotaxime ($m/z$ = 456.0642).

9.56 and RE% ranged from 1.86 to 11.11% over the three different concentrations. Inter-day RSD% ranged from 2.71 to 4.90% and RE% ranged from 2.83 to 9.49%.

There is no interference $m/z$ peak at both $m/z$ values of MTZ and the IS from the five different injections of the blank (ionization solvent), hence the method is selective. A high degree of selectivity was also examined in the validity of the developed nano-ESI-MS/MS method to determine MTZ in tablets as there was no interference arising from the commonly used additives. Experimental results after analysis of a 500 ng ml$^{-1}$ preparation offered good recovery of 104.11% with 3.44% RSD, making the present technique promising for quality control studies, as documented in table 3. When using the ionization solvent for preparation of MTZ standards and samples, with the aid of an IS, the matrix effects are not taken into account for validation of the method because they do not affect reproducibility or linearity of MTZ.

### 3.5.4. Detection of MTZ in human urine

When MTZ is prescribed for the treatment of trichomoniasis, it is unusual to discontinue treatment; however, it is possible that the patient may not take the medicine, and the doctor may wonder whether the medication has been properly absorbed or not. MTZ is also believed to reduce the desire for alcohol in alcoholism [48]. Here, patient cooperation may be questionable, and a question arises as to whether or not the drug is taken. Therefore, it may seem useful to determine whether MTZ has been taken and whether it has been absorbed or not. Several LC-MS methods can easily study the presence of MTZ in blood or in urine, but these cannot be carried out without sample pre-treatment procedures [49,50]. In the absence of a blood sample, the simple and quick detection of MTZ in urine is of practical interest upon applying the direct MS technique. Thus, the developed direct NS-ESI-MS/MS method was applied to detect the analyte in random human clinical urine samples after oral administration of the MTZ or MTZ with other co-administered prescribed drugs. The obtained results were compared with the medical records of the patients to verify the reliability of the method. Figure 6 reports the results obtained by analysing the drug-free human urine sample (figure 6a), spiked drug-free urine sample (figure 6b) and real sample containing MTZ alone (figure 6c). Figure 7 reports the result obtained by analysing the real sample containing MTZ with other co-administered drugs (vitamin B3, omeprazole, lactulose and cefotaxime). The results showed that direct detection of MTZ is feasible despite the presence of other matrix interfering compounds or in the presence of potentially co-administered drugs.

## 4. Conclusion

A rapid, sensitive and selective direct NS-ESI-MS/MS technique has been developed and validated to quantify MTZ in tablet dosage forms. The direct NS-ESI-MS/MS system developed in the present study can provide an alternative to conventional chromatographic techniques and be very promising to various applications, based on the very low sample volume and short runtime. The direct NS-ESI-MS/MS measurement exhibits a strong ionization technique and provides high resolution, which can provide easy-to-interpret mass spectra for MTZ and other pharmaceuticals. The developed method was also designed for continuous detection of drug intake of patients via screening of the drug in urine samples. Urine samples collected from volunteers were analysed directly without any extraction process. The obtained results were in agreement with the medical records of the volunteers which, in turn, demonstrated the reliability of the method. The results also reveal that direct detection of MTZ in human urine is feasible with direct NS-ESI-MS/MS despite the presence of matrix interfering compounds or other co-administered drugs. Thus, the developed method can be used in clinical practices to detect MTZ or other pharmaceuticals in urine samples in a short time.

Ethics. All volunteers gave informed consent prior to donating urine.

Data accessibility. The datasets supporting this article have been uploaded in the Dryad Digital Repository: https://dx.doi.org/10.5061/dryad.778340g [51].

Authors' contributions. G.H. and S.E. were involved in conceptualization; T.M. contributed to methodology; W.Z. and H.E.-G. contributed to software; S.A., W.Z. and H.E.-G. validated; S.A. contributed to formal Analysis.; W.Z., H.E.-G. and R.A. contributed to investigation; W.Z. and H.E.-G. took care of resources; S.A., W.Z. and H.E.-G. contributed to data duration; writing—original draft preparation was done by S.A.; writing—review and editing were done by W.Z., H.E.-G. and S.E.; S.E. supervised.

Competing interests. We declare we have no competing interests.

Funding. We received no funding for this study.

Acknowledgements. S.A. acknowledges the support of RIKEN, Quantitative Biology Center—Japan, for the Internship programme and she is grateful to all technical staffs of the Laboratory for Single Cell Mass Spectrometry for their extensive help during her internship at RIKEN.

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
