## [Reviewer comments · Royal Society Open Science]

Review History

RSOS-191336.R0 (Original submission)

Review form: Reviewer 1

Is the manuscript scientifically sound in its present form?

Yes

Are the interpretations and conclusions justified by the results?

Yes

Is the language acceptable?

Yes

Do you have any ethical concerns with this paper?

No

Have you any concerns about statistical analyses in this paper?

No

Recommendation?

Accept with minor revision (please list in comments)

Comments to the Author(s)

The manuscript "Direct nano-electrospray ionization tandem mass spectrometry for quantification and identification of metronidazole in its dosage form and human urine" provides a well-established methodology protocols for ease of quantification and detection of MTZ in urine sample (especially in a clinical setting).

The paper is well suited in the Royal Society scope and I recommend it to be published with minor grammatical corrections:

- 1- Figures should be clear (m/z values are not very clear)
- 2- Figure 5 should have labels (a), (b), and (c) for each spectrum

Review form: Reviewer 2**Is the manuscript scientifically sound in its present form?**

Yes

Are the interpretations and conclusions justified by the results?

Yes

Is the language acceptable?

Yes

Do you have any ethical concerns with this paper?

No

Have you any concerns about statistical analyses in this paper?

No

Recommendation?

Accept with minor revision (please list in comments)

Comments to the Author(s)

In this work, direct nano-electrospray ionization-tandem mass spectrometry (NS-ESI-MS/MS) was developed and validated to quantify MTZ in tablet dosage forms. Direct NS-ESI-MS/MS was also designed for continuous detection of drug intake of patients via screening of the drug in urine samples despite the presence of matrix interfering compounds or other co-administered drugs. Obviously, direct MS is of great importance due to combining the benefits of the detection and quantification without the separation technique in coupling with MS. Both of the originality and experimental data are good. In general term, the present study has enough quality to be published in Royal Society Open Science but some minor revisions are necessary before it may be accepted.

- 1- Trichomoniasis vaginalis should be italicized (p1, line 46)

2- Standard Solutions and calibration

According to the results in Table 1 and linearity, the working standard solution range should be from 0.25-2500 µg/mL and subsequently, the calibration standard solutions ranging from 2.5 to 25000 ng/mL

3- The identity of parts a, b and c in figure 5 should be indicated.

4- Annotation of fragments should be corrected (underscore of molecule numbers) (H₂O and NH₃)

5- Detection of MTZ in human urine

5.1. "MTZ is also believed to reduce the desire for alcohol in alcoholism"; "Several LC-MS methods can easily study the presence of MTZ in blood or in urine but these cannot be carried out without sample pretreatment procedures"...References should be given

5.2. Figures 6 reported only the results obtained by analyzing the drug-free human urine sample (Figure 6a), spiked drug-free urine sample (Figure 6b) and real sample containing MTZ alone (Figure 6c), while Figure 7 reported the result obtained by analyzing real sample containing MTZ with other co-administered drugs

6- Data Extraction and Interpretation

There should be a singular formatting style when recalling figures. When referring to figure 4, parts a and b should appear in the text.

7- In section 3.3, the last sentence" The goal of calibration was to calibrate the instrument to detect the correct mass" is not needed

8- In section 3.5, the last two sentences should be deleted and instead should state that the same method for dosage method was applied.

9- Figures 1, 2, and 3 should have the top heading removed.

Decision letter (RSOS-191336.R0)

09-Sep-2019

Dear Dr Amer:

Title: Direct nano-electrospray ionization tandem mass spectrometry for quantification and identification of metronidazole in its dosage form and human urine
Manuscript ID: RSOS-191336

Thank you for submitting the above manuscript to Royal Society Open Science. On behalf of the Editors and the Royal Society of Chemistry, I am pleased to inform you that your manuscript will be accepted for publication in Royal Society Open Science subject to minor revision in accordance with the referee suggestions. Please find the reviewers' comments at the end of this email.

The reviewers and handling editors have recommended publication, but also suggest some minor revisions to your manuscript. Therefore, I invite you to respond to the comments and revise your manuscript.

Because the schedule for publication is very tight, it is a condition of publication that you submit the revised version of your manuscript before 18-Sep-2019. Please note that the revision deadline will expire at 00.00am on this date. If you do not think you will be able to meet this date please let me know immediately.

Best wishes,
Dr Laura Smith
Publishing Editor, Journals

Royal Society of Chemistry
Thomas Graham House

Science Park, Milton Road
Cambridge, CB4 0WF
Royal Society Open Science - Chemistry Editorial Office

RSC Associate Editor:
Comments to the Author:
(There are no comments.)

RSC Subject Editor:
Comments to the Author:
(There are no comments.)

Reviewer comments to Author:
Reviewer: 1

Comments to the Author(s)

The manuscript "Direct nano-electrospray ionization tandem mass spectrometry for quantification and identification of metronidazole in its dosage form and human urine" provides a well-established methodology protocols for ease of quantification and detection of MTZ in urine sample (especially in a clinical setting).

The paper is well suited in the Royal Society scope and I recommend it to be published with minor grammatical corrections:

- 1- Figures should be clear (m/z values are not very clear)
- 2- Figure 5 should have labels (a), (b), and (c) for each spectrum

Reviewer: 2

Comments to the Author(s)

In this work, direct nano-electrospray ionization-tandem mass spectrometry (NS-ESI-MS/MS) was developed and validated to quantify MTZ in tablet dosage forms. Direct NS-ESI-MS/MS was also designed for continuous detection of drug intake of patients via screening of the drug in urine samples despite the presence of matrix interfering compounds or other co-administered drugs. Obviously, direct MS is of great importance due to combining the benefits of the detection and quantification without the separation technique in coupling with MS. Both of the originality and experimental data are good. In general term, the present study has enough quality to be published in Royal Society Open Science but some minor revisions are necessary before it may be accepted.

- 1- *Trichomoniasis vaginalis* should be italicized (p1, line 46)
- 2- Standard Solutions and calibration

According to the results in Table 1 and linearity, the working standard solution range should be from 0.25-2500 µg/mL and subsequently, the calibration standard solutions ranging from 2.5 to 25000 ng/mL

3- The identity of parts a, b and c in figure 5 should be indicated.

4- Annotation of fragments should be corrected (underscore of molecule numbers) (H₂O and NH₃)

5- Detection of MTZ in human urine

5.1. "MTZ is also believed to reduce the desire for alcohol in alcoholism"; "Several LC-MS methods can easily study the presence of MTZ in blood or in urine but these cannot be carried out without sample pretreatment procedures"...References should be given

5.2. Figures 6 reported only the results obtained by analyzing the drug-free human urine sample (Figure 6a), spiked drug-free urine sample (Figure 6b) and real sample containing MTZ alone (Figure 6c), while Figure 7 reported the result obtained by analyzing real sample containing MTZ with other co-administered drugs

6- Data Extraction and Interpretation

There should be a singular formatting style when recalling figures. When referring to figure 4, parts a and b should appear in the text.

7- In section 3.3, the last sentence "The goal of calibration was to calibrate the instrument to detect the correct mass" is not needed

8- In section 3.5, the last two sentences should be deleted and instead should state that the same method for dosage method was applied.

9- Figures 1, 2, and 3 should have the top heading removed.

Author's Response to Decision Letter for (RSOS-191336.R0)

See Appendix A.

Decision letter (RSOS-191336.R1)

04-Oct-2019

Dear Dr Amer:

Title: Direct nano-electrospray ionization tandem mass spectrometry for quantification and identification of metronidazole in its dosage form and human urine

Manuscript ID: RSOS-191336.R1

It is a pleasure to accept your manuscript in its current form for publication in Royal Society

Open Science. The chemistry content of Royal Society Open Science is published in collaboration with the Royal Society of Chemistry.

RSC Associate Editor
Comments to the Author:
(There are no comments.)

Reviewer(s)' Comments to Author:

Appendix A

Response to reviewer comments

Reviewer: 1

1- Figures should be clear (m/z values are not very clear)

1- All figures were redone and uploaded in higher quality to be read clearly.

2- Figure 5 should have labels (a), (b), and (c) for each spectrum

2- Labels have been added to Figure 5.

Reviewer: 2

1- Trichomoniasis vaginalis should be italicized (p1, line 46)

1- Trichomoniasis vaginalis was italicized in the revised text.

2- Standard Solutions and calibration

According to the results in Table 1 and linearity, the working standard solution range should be from 0.25-2500 µg/mL and subsequently, the calibration standard solutions ranging from 2.5 to 25000 ng/mL

2- Concentrations were corrected in the text to match the results of Table 1.

3- The identity of parts a, b and c in figure 5 should be indicated.

3- Labels have been added to Figure 5.

4- Annotation of fragments should be corrected (underscore of molecule numbers) (H₂O and NH₃)

4- Molecule annotation was corrected in the text.

5- Detection of MTZ in human urine

5.1. "MTZ is also believed to reduce the desire for alcohol in alcoholism"; "Several LC-MS methods can easily study the presence of MTZ in blood or in urine but these cannot be carried out without sample pretreatment procedures"...References should be given

5.1. References [48-50] were added in the text.

5.2. Figures 6 reported only the results obtained by analyzing the drug-free human urine sample (Figure 6a), spiked drug-free urine sample (Figure 6b) and real sample containing MTZ alone (Figure 6c), while Figure 7 reported the result obtained by analyzing real sample containing MTZ with other co-administered drugs

5.2- Figure referral was revised in the text.

6- Data Extraction and Interpretation

There should be a singular formatting style when recalling figures. When referring to figure 4, parts a and b should appear in the text.

6- Text was revised in Data Extraction and Interpretation to properly refer to Figure 4(a) and 4(b).

7- In section 3.3, the last sentence "The goal of calibration was to calibrate the instrument to detect the correct mass" is not needed

7- This sentence was removed from the revised text.

8- In section 3.5, the last two sentences should be deleted and instead should state that the same method for dosage method was applied.

8- The last sentence in section 3.5 was revised as follows:

"Urine samples were loaded into the nanospray capillaries and processed as mentioned under instrumentation. "

9- Figures 1, 2, and 3 should have the top heading removed.

9- Figures 1,2, and 3 were edited accordingly.

10- Figures 1, 2, and 3 should have the top heading removed.

10- Top heading of figures 1, 2, and 3 were removed